# Computed Tomographic Assessment of Pituitary Gland Dimensions in Domestic Short-Haired Cats

**DOI:** 10.3390/ani13121935

**Published:** 2023-06-09

**Authors:** Dario Costanza, Pierpaolo Coluccia, Luigi Auletta, Erica Castiello, Luigi Navas, Adelaide Greco, Leonardo Meomartino

**Affiliations:** 1Interdepartmental Center of Veterinary Radiology, University of Napoli “Federico II”, Via Federico Delpino 1, 80137 Napoli, Italy; dario.costanza@unina.it (D.C.); pierpaolo.coluccia@unina.it (P.C.); erica.castiello@unina.it (E.C.); leonardo.meomartino@unina.it (L.M.); 2Department of Veterinary Medicine and Animal Sciences (DIVAS), University of Milano, Via dell’Università 6, 26900 Lodi, Italy; luigi.auletta@unimi.it; 3Department of Veterinary Medicine and Animal Production, University of Napoli “Federico II”, Via Federico Delpino 1, 80137 Napoli, Italy; luigi.navas@unina.it

**Keywords:** acromegaly, adenoma, diabetes, feline, hypophysis, microadenoma

## Abstract

**Simple Summary:**

The pituitary gland is crucial in regulating metabolic processes. Tumors in this gland can be subtle, meaning that establishing reference values for the pituitary dimensions is pivotal. Computed tomography is commonly used to identify pituitary alterations, plan surgery or radiation therapy, and monitor treatment responses. This study aimed to determine the normal pituitary size, and the pituitary-to-brain ratio, in a group of domestic short-haired cats by computed tomography. The study also aimed to explore the correlations between body weight, age, sex, and pituitary dimensions, and assess the inter- and intra-agreement between operators in measuring pituitary dimensions. The study showed that the normal range for pituitary dimensions is wider than previously reported. The study also showed a low correlation between body weight, age, and pituitary dimensions. The intra-operator agreement in measuring pituitary dimensions was good/excellent, but the inter-operator agreement was moderate/good, likely due to differences in expertise. The reference values obtained from this study could help in better evaluating the pituitary gland in domestic short-haired cats with suspected pituitary neoplastic lesions.

**Abstract:**

The detection of subtle changes in the pituitary dimensions has relevant clinical implications. In cats, a few studies have established the cut-off values of the pituitary gland’s dimensions using small and inhomogeneous samples. The aims of this study were: to determine by computed tomography (CT) the pituitary linear dimensions and the pituitary-to-brain (P:B) ratio in a sample of domestic short-haired (DSH) cats; to assess the effects of sex, age, and weight on pituitary dimensions; and to evaluate the inter- and intra-observer agreement for such measurements. All skull CTs of DSH cats performed over four years using a multidetector CT and a standardized protocol were retrospectively reviewed. The exclusion criteria were: clinical, laboratory, or CT alterations of the pituitary gland, brain diseases, fractures of the neurocranium, and diabetes. The pituitary dimensions and brain area were assessed by two different observers using multiplanar reconstructions and automated segmentation tools. Fifty-one cats were included in the final sample. The intraclass correlation coefficients for intra- and inter-observer reliability were good/excellent, and moderate/good, respectively. No differences between sexes were detected, and negligible correlations were found between age and weight. According to this study, a pituitary gland with a height > 4 mm or a P:B ratio > 0.49 mm should be considered enlarged.

## 1. Introduction

The pituitary gland plays a pivotal role in regulating the endocrine system through the production, storage, and release of various hormones [1]. Pituitary tumors can originate from different cell lineages, and can be functional or non-functional [2,3,4,5,6]. The clinical signs associated with the neoplasm depend on the secretory proprieties, but also on the tumor’s size [2,4,6,7,8,9]. Indeed, non-functional tumors can also become clinically relevant when they enlarge enough to cause neurologic signs by the direct compression of other intracranial structures [3,4,5,6,10]. In dogs, corticotroph (ACTH-secreting) adenomas and adenocarcinomas responsible for pituitary-dependent hyperadrenocorticism prevail, while in cats, somatotroph (growth hormone (GH)-secreting) tumors are the most reported [5,6]. The excessive secretion of GH can result in chronic hypersomatotropism that can cause acromegaly and insulin resistance due to the concomitant increase in insulin-like growth factor-1 (IGF-1). Of note, cats affected by somatotroph pituitary tumors are often brought in for consultation for clinical signs related to poorly controlled diabetes, such as polyuria, polydipsia, and polyphagia, rather than for somatic changes due to acromegaly [10,11,12]. Somatotroph adenoma is the most reported pituitary tumor in middle-aged to older male cats. Domestic short-haired (DSH) cats and Maine Coons seem predisposed [10,11,12]. 

The detection of anatomical alterations of the pituitary gland is usually performed using magnetic resonance imaging (MRI) or computed tomography (CT) [7]. These imaging techniques are paramount for diagnosis, surgical planning, or radiotherapy [8]. In many cases, the diagnosis of pituitary macroadenoma is straightforward, since the pituitary mass dorsally protrudes from the sella turcica, and compresses the adjacent brain parenchyma, sometimes with associated neurological signs [7,8,9]. However, in the case of microadenomas, there are often only subtle changes in the pituitary size and contours that may not be visible on cross-sectional imaging (MRI, CT) [9,12]. Since pituitary tumors can alter the regular network of pituitary vessels and, consequently, the enhancement pattern, the pituitary gland was also evaluated using CT dynamic scans, in order to increase diagnostic accuracy in detecting pituitary microadenomas [13]. 

A few studies have attempted to establish the linear dimensions of the pituitary gland (height, width, and length) in cats using either MRI [14] or CT [13,15,16]. The pituitary height-to-brain ratio (P:B ratio) was introduced to address variation in pituitary gland size between dogs and cats of different sizes and breeds [15,16,17]. However, those studies performed in cats used third-generation CT devices [15] or thick MRI slices [14], and included small and inhomogeneous samples. More recently, a study aiming to evaluate possible differences in pituitary size between mesaticephalic and brachycephalic cats found a significant difference between the two skull morphotypes [16]. These results highlight the need to obtain reference measures of the pituitary gland according to the morphotype or, even better, according to the breed. To date, a single study, including only DSH cats, described the pituitary dimensions and P:B ratio [15]. However, in our clinical experience, cats without hematological, clinical, and CT findings of pituitary disease had pituitary dimensions outside the previously proposed cut-off values. Consequently, we hypothesized that the reference intervals for pituitary linear dimensions (height, length, and width) and the P:B ratio in DSH cats would be different from those previously reported. 

Therefore, the primary aim of the present study was to verify if the previously reported cut-off values for pituitary linear dimension and P:B ratio were adhered to in a larger sample of DSH cats and, eventually, to establish new reference values. The secondary objectives were to evaluate the influence of age, body weight, and sex on the pituitary dimensions and P:B ratio. Finally, to assess the reliability of, and variability in, the performing of the pituitary linear dimensions measurements and the obtaining of the P:B ratio, we evaluated the intra- and inter-observer agreement between two observers with different levels of expertise.

## 2. Materials and Methods

### 2.1. Selection and Description of Subjects

This single-center, retrospective, reference interval, intra- and inter-observer agreement study was approved by the Ethical Animal Care and Use Committee of the University of Napoli “Federico II” (Prot. N. PG/2023/0050567). The electronic clinical records and CT studies of cats referred to the Interdepartmental Centre of Veterinary Radiology of the University of Napoli “Federico II” in the set study period between September 2018 and October 2022 were retrieved from the picture archiving and communication system (dcm4chee-arc-light version 5.11.1, http://www.dcm4che.org, URL accessed on 2 May 2023). The inclusion criteria were the DSH breed, and the same CT unit and scanning protocol. Patients were excluded from the study if they had a clinical, laboratory, or CT final report referring to any alteration related to (a) pituitary gland disease, (b) neurologic signs different from vestibular symptoms, (c) intracranial lesions detected on CT, (d) fractures or conformational alterations of the neurocranium, (e) polyuria or polydipsia, (f) signs of acromegaly, (g) definitive diagnosis of diabetes mellitus (persistent hyperglycemia, glucosuria, increased level of serum fructosamine, and consistent clinical signs). 

### 2.2. CT Scan Protocol

All patients were positioned in sternal recumbency within a radiolucent polyurethane vacuum immobilization mattress (Vacuumat, Génia, St. Hilaire de Chaléons, France), and with the forelimbs pulled caudally along the thorax, under general anesthesia (the anesthetic protocol adopted varied depending on the decision of the anesthesiologist in charge). Computed tomography studies were obtained using a 16-slice multidetector computed tomography (MDCT) unit (BrightSpeed, General Electric Healthcare, Milwaukee, WI, USA). The acquisition protocol was: helical mode; slice thickness 1.25 mm; pitch 0.9375:1; 120 kVp, 160–200 mA, 1 s tube rotation speed; soft tissue and bone reconstruction algorithms (General Electric proprietary “standard” and “bone” filters). All the patients received a standardized intravenous dose (740 mgI/kg, i.e., 2 mL/kg) of contrast media (Iopamidol, Iopamiro 370 mgI/mL, Bracco Imaging s.p.a., Milano, Italy) using an infusion rate of 1 mL/s followed by a 5 mL saline flush through a double-barrel power injector (EmpowerCTA+, Bracco Imaging s.p.a., Milano, Italy). Postcontrast images were acquired after a fixed delay of 60 s. 

### 2.3. Data Recording and Image Analysis 

A veterinary radiologist (L.M., full Professor of Veterinary Radiology with a Ph.D. and >25 years of experience) performed a preliminary evaluation to include or exclude each cat from the definitive sample group. The CT studies were excluded if: (a) the postcontrast series of the skull acquired with soft tissue reconstruction algorithm was not available, (b) the quality of the images was deemed inadequate for a correct interpretation, or (c) the presence of beam-hardening artifacts was detected, preventing a correct evaluation of the pituitary gland. The same author anonymized all the CT studies before submitting them to two observers, who reviewed the images using commercial DICOM viewer software (Philips Extended Brilliance Workspace v. 4.5.5, Philips Medical System Nederland B.V., Best, The Netherlands). Observer 1 (D.C., a third-year Ph.D. student in Veterinary Diagnostic Imaging) and Observer 2 (P.C., a veterinarian with two years of expertise in CT imaging) were blinded regarding the clinical data and reasons for the CT examination. All the measurements were performed once by each observer, independently of, and blinded to, the results reported by the other observer. A pituitary gland measurement method was established prior to the analysis by two authors (D.C. and L.M.), and recorded in a portable document file. Pituitary linear dimensions (height, length, width) expressed in millimeters (mm) were measured using electronic calipers on postcontrast images displayed using a standardized window [window width (WW): 450, window level (WL): 200], although WW and WL could be manually adjusted by the observers if they deemed it necessary. Measurements were made on multiplanar reconstruction (MPR) images in order to obtain the best visualization of the pituitary gland, and avoid interpreting the dorsum sellae as a pituitary mass [9]. The pituitary height (PH) was measured at the level of the pituitary fossa, perpendicular to the basisphenoid bone, where the maximal pituitary height was visible, both on the transverse plane (PHT) (Figure 1A) and on the sagittal plane (PHS) (Figure 1B). The pituitary length (PL) was measured on the sagittal plane where the maximal length was visible, parallel to the basisphenoid bone (Figure 1B). The pituitary width (PW) was determined on the transverse plane at the point of maximal width of the gland (Figure 1C). The brain area (BA) expressed in mm^2^ was measured on the same slice of the PHT, using an automated segmentation tool (Figure 1D). When deemed necessary, in order to better delineate the brain edges, the observer used the bone reconstruction algorithm with the associated high-contrast window (WW = 2000, WL = 800). All the data were reported in an electronic spreadsheet (Microsoft Excel version 16.52 2021, Microsoft Corp. Redmond, WA, USA), and the P:B ratio was automatically computed for each cat as follows: P:B ratio = [PHT(mm) × 100/BA(mm^2^)] [17]. To assess the intra-observer agreement, the two observers repeated the measurements two months after the first evaluation, on a smaller sample of thirty re-anonymized and re-randomly selected CT exams. The sex, neutering status, weight (in kilograms), and age (in months) were recorded for each cat included in the final sample group.

### 2.4. Statistical Analysis 

Statistical analyses were performed by one of the authors (L.A., a former researcher with a Ph.D. and >10 years of experience and training in statistics), using commercial software (JMP Pro, v. 16.0, SAS Institute, Cary, NC, USA; MedCalc version 19.2.6, MedCalc Software Ltd., Ostend, Belgium; IBM SPSS, v. 26.0, IBM, Armonk, NY, USA). The normality of data was assessed using the Shapiro–Wilk W test. Continuous data were reported as mean ± standard deviation (SD) or median (range), depending on the distribution. 

The reference range for each measurement was calculated according to the American Society for Veterinary Clinical Pathology (ASCVP) guidelines for reference intervals [18]. In brief, outliers were automatically identified according to Reed et al. [19]. Then, data distribution was tested automatically using the Shapiro–Wilk W test. The reference lower and upper limits, and the corresponding 90% confidence intervals (CIs), were then calculated, employing the robust method following Clinical Laboratory and Standards Institute (CLSI) recommendations (CLSI C28-A3), with bootstrapping (1000 iterations) [20]. A Bland–Altman plot was used to explore the differences between PHS and PHT. The bias and 95% limits of agreement were calculated. Correlations between the measurements and body weight were tested with Pearson’s correlation coefficient (r), whereas those between the measurements and age were tested using Spearman’s rank correlation coefficient (r_s_). Finally, differences between males and females were tested with a pooled Student’s t-test, with Welch’s correction, as variance results were significantly different at the F test (PL, PW, BA, and P:B ratio), or a Mann–Whitney U test (PHT and PHS), according to sex distribution and non-considering of the neutering status. No partitioning into subclasses based on age, body weight, or sex was applied in the reference range calculations, due to the lack of correlation with the measurements. 

For inter-observer reliability, measurements belonging to all the patients were included in the analysis, while for intra-observer reliability, only the measurements of the smaller sample were considered. For inter- and intra-observer agreement, a two-way mixed-effects intraclass correlation coefficient (ICC), for single measurement or single observer accordingly, and absolute agreement, were calculated; the relative 95% CIs were also calculated. The ICC was categorized according to Koo and Li [21]. Based on the reliability analysis results, all intra- and inter-observer measurements with an ICC > 0.80 were averaged and subsequently analyzed. For measurements that did not reach this value, the first measurements of Observer 1, considered the most experienced, were used. In all analyses, *p* < 0.05 was considered statistically significant.

## 3. Results

Fifty-nine CT studies performed in the set period met the inclusion criteria. After the preliminary review, eight cases were excluded for the following reasons: brain neoplasia (*n* = 4), deformed cranium (*n* = 2), and lack of postcontrast images (*n* = 2). Fifty-one DSH cats were included in the final sample; they were 4 intact females (8%), 26 (51%) spayed females, 7 (14%) intact males, and 14 (27%) castrated males. The median age was 72 months (range 2–180), and the mean weight was 4.6 ± 1.5 kg.

In all the cats included in the final sample, the pituitary gland was distinguishable from the adjacent structures with a good contrast enhancement. All the predetermined measurements were obtained. In three cases, it was challenging to identify the pituitary edges with absolute precision, due to the pituitary gland’s small size. The automated segmentation tool allowed for a rapid delimitation of the BA in all cases. 

The results indicated that all the measurements described were normally distributed, and no outliers were detected in any of them. The reference intervals, mean ± SD, and range (minimum to maximum) for linear measurements and P:B ratio are reported in Table 1.

Regarding the pituitary height, the mean (±SD) PHT was 2.94 ± 0.52 mm and the PHS 2.95 ± 0.55 mm, and the Bland–Altman evaluation resulted in a bias of −0.01 and limits of agreement of −0.79–0.76 (Figure 2). 

All measurements showed negligible-to-low correlation with body weight, whereas only PHS and PL showed a low correlation with age. Correlation coefficients and relative *p*-values are summarized in Table 2. No differences between sexes were detected for any measurement (PHT, *p* = 0.40; PHS, *p* = 0.68; PL, *p* = 0.31; PW, *p* = 0.21; BA, *p* = 0.30; P:B ratio, *p* = 0.94).

The results for the inter-observer agreement are summarized in Table 3. All measurements were within the moderate reliability class, except for BA, where good reliability was detected. 

The results for the intra-observer agreement are summarized in Table 4. Reliability was good for all measurements from Observer 1 except for PL, which results indicated as moderate; on the other hand, results indicated moderate reliability for all measurements from Observer 2 except for BA, which results indicated as excellent.

## 4. Discussion

The primary aim of the present study was to verify if the previously reported cut-off values for pituitary linear dimension and P:B ratio were adhered to in a larger sample of DSH cats and, eventually, to establish new reference values. The obtained values partly differ from those already reported in the literature [13,14,15,16]. This discrepancy might be related to the differences in sample size. Effectively, to obtain reliable reference values, a sample size of 120 healthy patients should be collected [18,19,22]; nonetheless, robust statistical methods have been developed for sample sizes of less than 120 and more than 40, such as that used in our work; these methods allow the calculation of the reference limits and 90% CIs [18]. In this study, the reference values for the pituitary linear dimensions and the P:B ratio were found according to the guidelines of the ASCVP [18]. The calculation of the reference limits and 90% CIs for sample sizes less than 40 has been discouraged [18,22]. 

Another possible cause of discrepancy between reported values and those from the current study may be the use of different imaging techniques (CT vs. MRI), different CT devices, or scan protocols. It is well known that MRI has a higher soft-tissue contrast compared to CT, which might give more accurate and reproducible results [7,23]. On the other hand, older MRI devices might not have enough field strength or precise electronic calipers in the range of millimeters [11]. Differences between CT devices, scan protocols [24,25], or even contrast dosage and infusion rate [26] can all affect image quality and, consequently, the correctness, reproducibility, and accuracy of the measurements. In the current study, all the CT studies were performed using an MDCT unit that, unlike older single-slice third-generation units, provides high spatial resolution and real isotropic reconstructions [25]. These technical aspects gave true MPRs, and allowed the precise assessment of the pituitary gland [27], providing images free from ring artifacts, common when older third-generation CT units are used [28]. 

Moreover, we selected only DSH cats, reducing eventual inter-breed variability. A significant difference in pituitary-gland linear measures between brachycephalic and mesaticephalic cats has already been demonstrated [16]; therefore, it is reasonable to hypothesize that other factors, e.g., the breed, might have some influence on these measures. Larger samples of cats stratified by breed might confirm this hypothesis. 

In other studies [11,14], both PHT and PHS have been reported, whereas Nadimi et al. reported only the PHT [15]. In the current study, the Bland–Altman plot showed minimal bias (−0.01) between PHT and PHS. This finding represents an indirect validation of the accuracy of the measurements reported, further confirmed by the differences in all the reported statistics (mean, SD, range, reference interval, and 90% CIs) that deviate from each other in the order of 0.1 mm. Therefore, it is the authors’ opinion that both the transversal and the sagittal scans can be used interchangeably to determine the pituitary height. 

One of the secondary objectives of this work was to evaluate the influence of age, body weight, and sex on the pituitary measurements, and also to evaluate if any sub-group had to be generated. All the linear measurements had a negligible/low correlation with body weight, and only PHS and PL had a low correlation with age. In a previous study, performed on fifteen cats [15], ten of which were DSH, the authors found a significant correlation between pituitary dimensions and body weight; however, the degree of this correlation was not reported. Additionally, considering only the DSH cats, the authors reported a significant difference between those weighing < 3 kg, and those above that body weight [15]. In our study, although body weight was positively correlated with all pituitary dimensions, the degree of correlation was negligible/low, in partial agreement with other studies [14,16]. Therefore, we did not consider it worth creating subgroups based on body weight. Further studies, with larger sample sizes, should explore this association. Nonetheless, in the authors’ opinion, body weight per se might not be the best variable to be considered. Perhaps, different body metrics, e.g., the body surface, body condition score, etc., could act as a better predictor of differences in pituitary gland dimensions. However, the P:B ratio might eliminate any influence of body weight and dimensions.

In humans, the progressive growth of the pituitary gland during puberty is well-known [29,30], as is the small size of the pituitary gland in elderly people [31]. In cats, Häußler et al. [16] found a positive correlation between age and pituitary height, width and length in the sub-group of brachycephalic cats, but 25% of them were under 15 months old. In contrast, this correlation was not explored in the mesaticephalic group. In our sample, 11 cats (21%) were under 15 months old, but we detected only a low correlation between PHS and PL, and age. Consequently, no conclusions can be drawn about the different results reported. Further studies should be designed to explore whether, in the feline species, a pattern of pituitary growth and late reduction in size, similar to that in human beings, exists. 

Finally, no correlation was found between gender and pituitary dimensions. This finding agrees with a previous study [15], but partially disagrees with that of Häußler et al. [16], in which male cats demonstrated a significantly larger pituitary width compared to female cats. However, the authors did not show the correlation coefficient, and significance was reached only in the mesaticephalic sub-group. In our study, the high number of neutered females and castrated males, which composed more than two-thirds of the whole sample (78%), might have introduced a bias, even if the neutering status has been reported to have had no influence on the pituitary dimensions [13]. Again, further studies with homogeneous-gender, larger groups are needed to clarify the influence of gender and neutering on the pituitary gland dimensions. 

The other secondary objective was the evaluation of the intra- and inter-observer agreement. The results of the ICC for the intra-observer agreement suggest that it was influenced mainly by the experience of the operator. Indeed, Observer 1, considered the most experienced, showed good reliability and, except for the PL and BA, displayed higher ICC values compared to Observer 2. The ICC for inter-observer agreement revealed a moderate/good agreement between the observers. In a previous study, assessing the intra- and inter-observer agreement accuracy and reproducibility of CT measurements of the pituitary gland in dogs using a phantom model, the authors found an excellent level of agreement for the PH and P:B ratio between the observers [32]. However, a systematic and significant difference was present between them. The authors concluded that due to this systematic variation, intra- and inter-patient comparisons should preferably be performed by the same observer [32]. In another study, Van Hoe et al. suggested a fundamental role of image-windowing in the manual measurement of small parts [33]. Accordingly, in the current study, notwithstanding the suggested pre-defined windowing, the moderate agreement between observers may rely on the operator’s experience and ability in setting the window level and width when it was considered necessary to better measure the pituitary gland. The ICC results for the BA indirectly support this hypothesis. Indeed, the BA was the measurement with the higher level of agreement between observers, and this is probably related to the automated segmentation tool used, thus eliminating the operator dependence. 

The main limitation of this study was the absence of an IGF-1 assay in cats included in the final sample. Consequently, it was impossible to categorically exclude acromegaly. However, none of the subjects had clinical, CT, or laboratory findings consistent with acromegaly or diabetes mellitus. Given the small number of laboratories that perform this analysis, and the relatively high cost, the assay for IGF-1 is not performed as part of the routine serum analysis panel at our institution, as it is reserved for cats with clinical suspicion of acromegaly. Another limitation was the absence of necropsy and, therefore, the possibility of ruling out pituitary lesions undetected by CT, and also of verifying the agreement between the actual ex vivo dimensions of the pituitary gland and those obtained on CT images. 

## 5. Conclusions

This study provides reference values for pituitary dimensions and P:B ratio in DSH cats. The reference values are wider than the mean values previously reported. The pituitary linear dimensions and P:B ratio exhibit a good intra-operator agreement, but a moderate inter-operator agreement, likely due to the different levels of expertise. Software that automatically defines the structures of interest may help to reduce this operator-dependent variability. In the sample analyzed, significant but negligible-to-low correlations were found between body weight and pituitary size, and between age and pituitary height and length. Hence, the actual effect of these variables on the pituitary gland remains questionable. In addition, no differences between genders were found. 

According to this study, a pituitary gland height > 4 mm, or a P:B ratio > 0.49 mm should be considered enlarged. The reference ranges obtained from this study may help assess pituitary gland size in DSH cats with suspected neoplastic lesions affecting the pituitary gland, in surgical or radiation therapy planning, and in monitoring responses to treatment.

## Figures and Tables

**Figure 1 animals-13-01935-f001:**
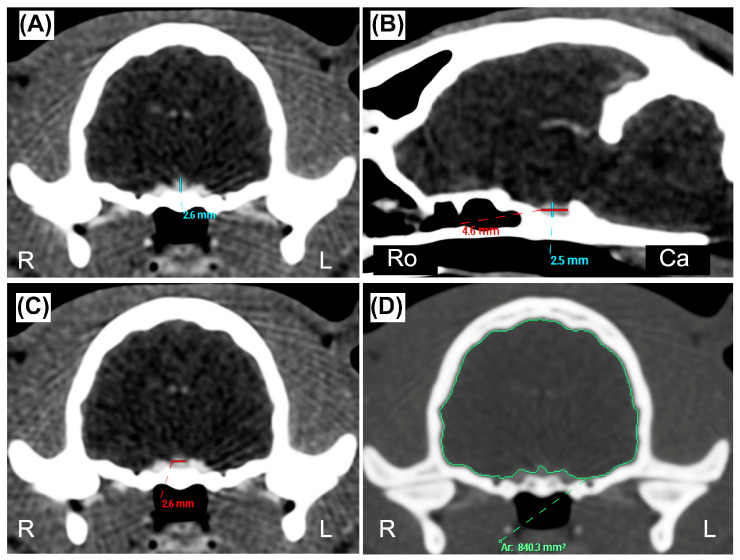
(**A**,**C**,**D**) Transverse and (**B**) sagittal postcontrast CT images of the skull of a twelve-month-old DSH cat. (**A**) Pituitary height (blue line) measured on the transverse plane (PHT) at the level of the pituitary fossa, perpendicular to the basisphenoid bone. (**B**) Pituitary height (blue line) measured on the sagittal plane (PHS) perpendicular to the basisphenoid bone, and pituitary length (PL, red line) measured where the maximal length of the pituitary gland was visible and parallel to the basisphenoid bone. (**C**) Pituitary width (red line) measured on the transverse plane (PW) at the point of maximal width of the gland. (**D**) Brain area (BA, green line) measured on the transverse plane, at the same level of the PTH, using an automated region of interest tool. (**A**–**C**) manually windowed to WW = 455, WL = 234; (**D**) manually windowed to WW = 2000, WL = 800. Abbreviations: Ca is Caudal; L is Left; R is Right; Ro is Rostral.

**Figure 2 animals-13-01935-f002:**
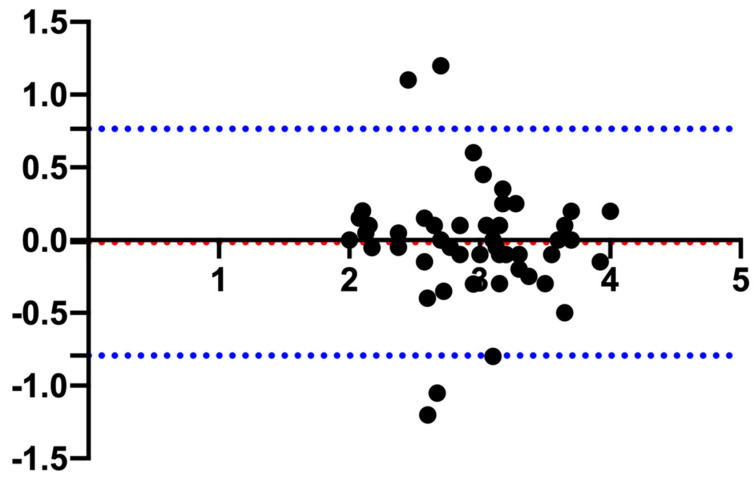
Bland–Altman plot comparing the measurements of the pituitary gland height measured in the transverse (PHT) and sagittal (PHS) plane. The *y*-axis shows the difference between the two measurements, and the *x*-axis shows the average. The blue dotted lines represent the 95% confidence intervals, and the red dotted line represents the bias.

**Table 1 animals-13-01935-t001:** Mean ± standard deviation, upper and lower limits of the reference value (and corresponding 90% CI) of the pituitary gland linear measurements, brain area, and pituitary-to-brain ratio.

Measurement	Mean (±SD)	Range (Min–Max)	Lower Limit (90% CI)	Upper Limit (90% CI)
PHT (mm)	2.94 (±0.52)	2.0–4.1	1.88 (1.68–2.10)	4.01 (3.81–4.20)
PHS (mm)	2.95 (±0.55)	1.9–4.0	1.84 (1.63–2.08)	4.10 (3.89–4.27)
PL (mm)	3.17 (±0.52)	2.0–4.4	2.13 (1.92–2.37)	4.23 (4.03–4.42)
PW (mm)	3.24 (±0.61)	2.1–4.8	1.92 (1.66–2.15)	4.41 (4.12–4.71)
BA (mm^2^)	789.02 (±61.85)	647.87–962.85	659.51 (634.51–687.61)	910.96 (883–938.01)
P:B ratio	0.37 (±0.06)	0.25–0.48	0.25 (0.23–0.28)	0.49 (0.47–0.51)

Abbreviations: CI is confidence interval; BA is brain area; Max is maximum; Min is minimum; P:B ratio is the pituitary gland height to brain area ratio; PHS is the maximal pituitary height on the sagittal plane; PHT is the maximal pituitary height on the transverse plane; PL is pituitary length; PW is pituitary width; SD is standard deviation.

**Table 2 animals-13-01935-t002:** Correlation coefficients of the measurements with body weight and age.

Measurement	Body Weight	*p*-Value	Age	*p*-Value
PHT (mm)	r = 0.39	0.004	r_s_ = 0.22	0.12
PHS (mm)	r = 0.30	0.032	r_s_ = 0.35	0.012
PL (mm)	r = 0.28	0.044	r_s_ = 0.36	0.009
PW (mm)	r = 0.35	0.011	r_s_ = 0.12	0.39
BA (mm^2^)	r = 0.29	0.036	r_s_ = 0.17	0.24
P:B ratio	r = 0.29	0.038	r_s_ = 0.14	0.33

Abbreviations: BA is brain area; P:B ratio is the pituitary gland height to brain area ratio; PHS is the maximal pituitary height on the sagittal plane; PHT is the maximal pituitary height on the transverse plane; PL is pituitary length; PW is pituitary width; r is Pearson’s correlation coefficient; r_s_ is Spearman’s rank correlation coefficient.

**Table 3 animals-13-01935-t003:** Intraclass correlation coefficient (ICC), relative 95% confidence intervals, and *p*-values for the inter-observer reliability test.

Measurement	ICC	95% CI	*p*-Value
PHT	0.69	0.38–0.84	<0.0001
PHS	0.58	0.37–0.74	<0.0001
PL	0.58	0.36–0.73	<0.0001
PW	0.66	0.46–0.79	<0.0001
BA	0.81	0.50–0.91	<0.0001

Abbreviations: BA is brain area; CI is confidence interval; ICC is the intraclass correlation coefficient; PHS is the pituitary height in the sagittal plane; PHT is the pituitary height in the transverse plane; PL is pituitary length; PW is pituitary width.

**Table 4 animals-13-01935-t004:** Intraclass correlation coefficient (ICC), relative 95% confidence intervals, and *p*-values for the intra-observer reliability test.

Measurement	Operator 1			Operator 2		
	ICC	95% CI	*p*-Value	ICC	95% CI	*p*-Value
PHT	0.81	0.63–0.91	<0.0001	0.66	0.17–0.86	<0.0001
PHS	0.87	0.71–0.94	<0.0001	0.60	−0.04–0.85	<0.0001
PL	0.51	0.19–0.73	0.001	0.70	0.10–0.89	<0.0001
PW	0.78	0.59–0.89	<0.0001	0.60	0.08–0.83	<0.0001
BA	0.82	0.66–0.91	<0.0001	0.92	0.84–0.96	<0.0001

Abbreviations: BA is brain area; CI is confidence interval; ICC is the intraclass correlation coefficient; PHS is the maximal pituitary height on the sagittal plane; PHT is the maximal pituitary height on the transverse plane; PL is pituitary length; PW is pituitary width.

## Data Availability

The data supporting the findings of this study are available from the corresponding author, upon reasonable request.

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
