# Peer review of "Computed Tomographic Assessment of Pituitary Gland Dimensions in Domestic Short-Haired Cats"

_animals, 2023, doi:10.3390/ani13121935_

Round 1
Reviewer 1 Report
The article is well organized and presents a god statistical approach to the data collected. In my opinion, a very good work was made.
The aim of the study is to determine the normal pituitary size and the pituitary to brain ratio in a in a group of domestic short-haired cats by computed tomography. The study also aimed to study the correlations between body weight, age, sex, and pituitary dimensions and to assess the inter- and intra-agreement between different operators.
The reference values obtained from the study can help evaluate the pituitary gland in domestic short-haired cats with suspected pituitary neoplastic lesions, aid in surgical or radiation therapy planning and monitor treatment response. Concerning the topic “diagnostic imaging of pituitary gland in small animals” the relevance of the paper is medium average.
Even if the relevance of the paper is medium average, the methodology of the study is excellent.
Tables and figure are clear and respect what explained in the text.
Author Response
Please see the attachement

Reviewer 2 Report
Dear Authors
Firstly: thanks for contributing to this interesting research topic
Secondly: after reviewing multiple papers of very poor quality (which cause A LOT of work for us reviewers) I am thrilled how well written and sound this paper is. Much appreciated.
I have some very minor suggestions:
35: As a suggestion I would rather write what the inclusion criteria were in the abstract
Line 43: I would strongly encourage you to clearly state in the abstract something similar to the line 390: According to this study, a pituitary gland height > 4 mm or a P:B ratio > 0.49 mm
should be considered enlarged.
That is what everyone will be looking for.
Line 54 to 70: I appreciate that a clinical connection needs to be made but I feel this is a bit too lengthy for a introduction. If possible please shorten or move to discussion.
78: may not be visible: on cross sectional imaging?
Line 86: it would be useful to clarify what a 3rd gen device is and why that would be a problem
Line 102: as a suggestion: It might be useful to say why it would be of value to answer the last question?
Line 117: Why definitive? And what makes a diagnosis of DM "definitive"?
Line 311 to 315: I do not think this paragraph is required and I would remove it. The following paragraph makes perfect sense without it.
There are some very minor things. Maybe just re-read (or have a native speaker read it)
Reviewer 3 Report
The paper is well thougth out, well written, with appropriate references.
summary: lines 26-28: repeated the concept from the first lines
Author Response
Reviewer 3: The paper is well thougth out, well written, with appropriate references. Summary: lines 26-28: repeated the concept from the first lines.
Authors: Thank you for your feedback. We have edited the last lines of the simple summary to avoid redundancy.
Reviewer 4 Report
|
Authors of the publication " Computed Tomographic Assessment of Pituitary Gland Dimensions in Domestic Short-Haired Cats" set themselves the task of verifying if the previously reported cut-off values for pituitary linear dimension and P:B ratio were respected in a larger sample of DSH cats and eventually, establish new reference values. Secondary objectives were to evaluate the influence of age, body weight and sex on pituitary dimensions and P:B ratio. Finally, the authors evaluated the intra- and inter-observer agreement among two observers with different levels of expertise. First, they performed a thorough review of the literature, from which, however, it appeared that in part the proposed study would be a repetition of earlier studies. However, in the paper a specialist with extensive experience in new generation diagnostic imaging equipment performed the study on a large group of cats, so the study methods can be described as modern and appropriate. The results obtained are described in detail and for the reader who is not a specialist in this type of diagnostics is understandable. In the discussion, the authors confronted the results obtained with the work of other authors, not avoiding the results of works on similar subjects, and giving in the final part of the discussion the limitations of the study and suggestions for further research in this area. Of particular interest seem to be the conclusions, which may not be of high scientific value, but may be of great applicative value for use by a practicing veterinary doctors. Out of reviewer's duty, I suggest removing, in my opinion, unnecessary self-citation of publication number 18.
|
|
Round 2
Reviewer 3 Report
All suggestion corrected with satisfaction